# Fungal Endophytes Isolated from *Elymus repens,* a Wild Relative of Barley, Have Potential for Biological Control of *Fusarium culmorum* and *Pyrenophora teres* in Barley

**DOI:** 10.3390/pathogens11101097

**Published:** 2022-09-25

**Authors:** Anna Kaja Høyer, Hans Jørgen Lyngs Jørgensen, Trevor Roland Hodkinson, Birgit Jensen

**Affiliations:** 1Botany, School of Natural Sciences, Trinity College Dublin, The University of Dublin, D02 PN40 Dubin, Ireland; 2Department of Plant and Environmental Sciences, Copenhagen Plant Science Centre, Faculty of Science, University of Copenhagen, DK-1871 Frederiksberg C, Denmark

**Keywords:** barley, biological control, *Elymus repens*, endophyte, *Fusarium culmorum*, *Periconia* sp., *Pyrenophora teres*

## Abstract

Twenty-four fungal endophytes, isolated from a wild relative of barley, *Elymus repens,* were screened in barley against an isolate of *Fusarium culmorum* and an isolate of *Pyrenophora teres* under controlled conditions. In all experiments, the endophytes were applied individually as seed dressings. Five endophytes could significantly reduce symptoms of *Fusarium culmorum* (*Periconia macrospinosa* E1 and E2*, Epicoccum nigrum* E4, *Leptodontidium* sp. E7 and *Slopeiomyces cylindrosporus* E18). In particular, treatment with *Periconia macrospinosa* E1 significantly reduced *Fusarium* symptoms on roots by 29–63% in two out of four experiments. Using, a *gfp* transformed isolate of *P. macrospinosa* E1, it was possible to show that the fungus was present on roots 14 days after sowing, coinciding with the disease scoring. To test for a potential systemic effect of the seed treatment, eight endophyte isolates were tested against the leaf pathogen *Pyrenophora teres.* Three isolates could significantly reduce symptoms of *P. teres* (Lasiosphaeriaceae sp. E10, Lindgomycetaceae sp. E13 and *Leptodontidium* sp. E16). Seed treatment with Lasiosphaeriaceae sp. E10 reduced net blotch leaf lesion coverage by 89%, in one out of three experiments. In conclusion, specific endophyte isolates exerted varying degrees of protection in the different experiments. Nevertheless, data suggest that endophytic strains from *E. repens* in a few cases are antagonistic against *F. culmorum* and *P. teres*, but otherwise remain neutral when introduced to a barley host in a controlled environment.

## 1. Introduction

Barley is the fourth most produced cereal crop in the world [1] and barley diseases result in great yield losses [2]. Some of the most important diseases in Northern Europe include leaf blotch caused by *Rhynchosporium graminicola,* Ramularia leaf spot (caused by *Ramularia collo-cygni*), Fusarium head blight (FHB, caused by *Fusarium* spp.) and net blotch (caused by *Pyrenophora teres*). FHB reduces the yield and quality of malt when producing beer and whiskey [3,4,5]. *Fusarium* spp. give early symptoms in roots and late symptoms occur in the heads [6,7]. Net blotch (*Pyrenophora teres*, [8,9]), causes symptoms on leaves and kernels [6].

Conventional agriculture relies on chemical inputs to control diseases, but foliar fungal pathogens are becoming tolerant to the fungicide treatments used [8] and soil-borne diseases are particularly challenging to target with chemical control [10]. In addition, it is anticipated that the use of some agrochemicals will be banned or restricted in the future [9] and in general, the European Union is promoting the sustainable use of pesticides and integrated pest management as part of their “Directive 2009/128/EC” [11,12]. Farmers need alternatives to chemical control and therefore there is growing interest in using microorganisms as biological control agents for plant diseases [13]. 

A constraint in the application of microorganisms for biological control is that they may have difficulties in persisting and/or remaining active when they are applied to the leaves, the seeds or the soil [14,15,16]. In contrast, endophytes, which are microorganisms living inside plants without causing symptoms of disease [17], are potentially favourable because the plant can protect the microorganism against unfavourable environmental conditions [18]. When protected within the plant, endophytes have the potential to provide control of several stresses without losing efficacy over the growing season. As an example, *Epichloë* spp. of forage grasses have been shown to provide both abiotic and biotic stress relief and the vertical transmission affirms the mutually beneficial symbiosis [19,20]. Some endophytes have been shown to protect plants against disease [21,22,23,24,25] and the use of endophytes against barley diseases was reviewed previously [2]. 

The objective of this study was to screen fungal root endophytes isolated from *Elymus repens* for their potential to control the foot and root rot pathogen *Fusarium culmorum* and the net blotch pathogen *Pyrenophora teres.* It was hypothesised that *E. repens* as a perennial wild relative of barley, would host endophytes that would be compatible for use in barley. The endophytes were dressed on barley seeds in all experiments. *F. culmorum* was also applied as a seed dressing and the two fungi were in close proximity to each other during the experiments. *Pyrenophora teres* was sprayed on leaves and thus a potential systemic effect was tested.

## 2. Results

### 2.1. Efficacy of Endophyte Seed Coating against Seed-Borne Fusarium culmorum 

The ability of 24 endophytes to control *F. culmorum* was tested in three experiments using a sand assay (exp. 1–3, Table 1, Appendix A) and three endophyte treatments were found to significantly reduce disease index, namely *Periconia macrospinosa* E1 with 63%, *Epicoccum nigrum* E4 with 38% and *Slopeiomyces cylindrosporus* E18 with 60% (Figure 1). When a subset of treatments was repeated (exp. 4–6, Appendix A), *P. macrospinosa* E1 was able to reduce disease symptoms in experiment 4 with 29%, but not in experiments 5 and 6. Treatment with *Epicoccum nigrum* E4 had no effect in experiments 4 and 5 (Figure 1).

Looking at disease incidence in the initial screening (Figure 2, exp. 1–3) resulted in the recording of five endophyte treatments that could significantly reduce the incidence of disease. Compared to the *Fusarium* treatment with 4% healthy plants in experiment 1, the following treatments had a significantly higher number of healthy plants: *Periconia macrospinosa* E1 at 52%, *Periconia macrospinosa* E2 at 25%, *Epicoccum nigrum* E4 at 38% and *Leptodontidium* sp. at 24%. In experiment 3, the treatment with *Slopeiomyces cylindrosporus* E18 had a significantly higher number of healthy plants at 75% compared to the *Fusarium* treatment at 43%. Interestingly, E1 and E2 were both isolates of *P. macrospinosa* cultured from the same plant (Table 2). However, when a subset of treatments was repeated there were no significant differences between endophyte treatments and the *F. culmorum* only control (exp. 4–6).

### 2.2. Root Colonisation by Periconia macrospinosa E1—Transformed with GFP

*Periconia macrospinosa* E1 was successfully transformed with GFP. The growth rates of the transformed strains were not different to the wild type (data not shown). The intensity of the emission was evaluated on a subjective scale from 0–5 for seven strains of E1gfp and, based on the intensity, E1gfp10 was chosen for the root colonisation experiment (Appendix A). 

*Periconia macrospinosa* E1gfp10 was present on roots 14 days after sowing (Appendix A) and it was present on 23 out of 24 root segments. However, colonies of *P. macrospinosa* E1gfp10 were not present in Petri dishes when competing against *F. culmorum* (Appendix A). 

### 2.3. Efficacy of Endophyte Seed Coating on Pyrenophora teres on Leaves

The ability of a subset of endophytes to control leaf infection of *Pyrenophora teres*, when the endophyte was dressed on barley seeds, was examined in experiments 7 to 9 (Table 1, Appendix A). In experiment 7, three endophyte isolates were able to reduce the percent lesion area of net blotch significantly, namely Lasiosphaeriaceae sp. E10 with 89%, Lindgomycetaceae sp. E13 with 59% and *Leptodontidium* sp. E16 with 58% (Figure 3). The three isolates were all cultured from different plants (Table 2). However, no significant differences in percent lesion area were found with Lasiosphaeriaceae sp. E10 in the replicate experiments 8 and 9. Treatment with *Periconia macrospinosa* E1 and *Epicoccum nigrum* E4 in experiments 8 and 9 also failed to reduce percent lesion area significantly despite having been successful against *F. culmorum* in previous experiments. 

## 3. Discussion

In this study, endophytes isolated from a wild relative of barley, *Elymus repens*, were tested against both a seed-borne pathogen (*Fusarium culmorum*) and a leaf pathogen (*Pyrenophora teres*) of barley. Since the endophytes were applied to seeds in all treatments, this implies that observed biocontrol efficacy probably was related to direct interaction in the case of seeds infested with *F. culmorum*, whereas, in the case of net blotch, a systemic effect could be expected as conidia of the pathogen were applied to the leaves. The 24 tested endophytes were originally isolated from eight different *E. repens* plants and interestingly, seven out of eight plants contributed an isolate that significantly reduced disease caused by either *Fusarium culmorum* or *Pyrenophora teres* in at least one experiment. 

To our knowledge, this is the first report of *Periconia macrospinosa* being tested against barley diseases. Two isolates of *P. macrospinosa,* E1 and E2, showed promising results against *F. culmorum* where they were able to reduce disease incidence in the initial experiment. Interestingly, the two isolates came from the same plant. Furthermore, isolate E1 also reduced disease severity in two out of four experiments. As an endophyte, *P. macrospinosa* is a species commonly associated with grasses including wheat [26,27,28] and was previously tested as a biocontrol agent against the take-all root disease (*Gaeumannomyces tritici*) of wheat [29]. However, they found that *P. macrospinosa* could not control take-all even when tested at very high concentrations. Instead of using seed coating, they inoculated the plants by mixing the inoculum of the pathogen and control agent into the soil.

The *Epicoccum nigrum* isolate E4 reduced *F. culmorum* disease severity and incidence in one out of three experiments. *Epicoccum* spp. have shown good biocontrol results in previous studies, both applied to seeds against soil-borne diseases [30,31] as well as sprayed for leaf-infecting [32] and post-harvest diseases [33]. El-Gremi et al. [31] showed that seed treatment with an *Epicoccum* isolate could significantly decrease disease symptoms of kernel black point caused by *Bipolaris sorokiniana*, *Alternaria alternata* and *Fusarium graminearum* in wheat under field conditions. Likewise, Hamza et al. [30] showed that *E. nigrum* could significantly reduce disease symptoms of late wilt of maize caused by *Magnaporthiopsis maydis.* El-Gremi et al. [31] applied the antagonist suspension to individual seeds at a concentration of 5×10^7^ spores/mL, whereas Hamza et al. [30] prepared the *E. nigrum* inoculum at a concentration of 3×10^5^ spores/mL. Other screening systems, more similar to field conditions, could have been used in our study to test the biocontrol agents against *F. culmorum.* However, Jensen et al. [34] found that the performance of their biocontrol agent in the same type of sand assay showed a high correlation to their results in subsequent field trials.

A biological control agent that would reduce symptoms of multiple diseases would be an asset to barley production. Thus, 8 of the 24 endophytes were tested against *Pyrenophora teres*. In two experiments, the promising candidates *Periconia macrospinosa* E1 and *Epicoccum nigrum* E4 with control potential against *F. culmorum* could not reduce disease caused by *P. teres.* The two endophytes would appear to have a more direct mode of action or simply only be antagonistic against *F. culmorum.* In contrast, seed treatment with Lasiosphaeriaceae sp. E10 reduced percent lesion area significantly in one out of three experiments, but Lasiosphaeriaceae sp. E10 had no effect against *Fusarium culmorum*. Furthermore, Lindgomycetaceae sp. E13 and *Leptodontidium* sp. E16 reduced net blotch symptoms significantly. However, when challenged with *Fusarium*, Lindgomycetaceae sp. E13 and *Leptodontidium* sp. E16 had no positive effects. In this case, the endophytes possibly had a systemic effect and were only antagonistic against *P. teres.* To elucidate whether the endophytes would also work against *P. teres* when in close contact, the endophytes could be sprayed onto the leaves before inoculation with the pathogen as done by Jensen et al. [35]. 

Only a limited number of studies have investigated the control of barley diseases using endophytes [2]. If the endophytic lifestyle of the biocontrol agent is the reason for the reduction of disease symptoms, then it is relevant to examine the colonisation of the isolate and certainly if the endophyte has been isolated from a different species than the crop [36]. In our experiments, *P. macrospinosa* conidia of the *gfp* transformant, applied to the seeds, resulted in root colonisation 14 days after sowing as shown by its growth on root segments incubated on agar. However, in the presence of *F. culmorum*, *P. macrospinosa* could not be re-isolated. Caution should be taken when interpreting in vitro growth experiments such as these since *F. culmorum* might be a better competitor on 1/5 PDA compared to *P. macrospinosa.* However, in the current experiment, seed coating with *P. macrospinosa* resulted in root colonisation 14 days later, demonstrating the ability of the isolate to thrive in the environment. Endophytic growth of the isolate was not tested in the current study. 

Of the successful endophytic control agents of barley diseases where the endophytic lifestyle in the target plants was confirmed, the endophytes were sourced from varying plant species including wild grasses such as *Ammophila arenaria* ssp. *australis* and *Corynephorus canescens* [21,37] as well as from two woody shrubs, *Prosopis juliflora* and *Zizyphus nummularia* [38,39]. It would appear logical that there is a potential for isolating effective control agents from *E. repens,* a grass found within the same tribe (Triticeae) as barley [40]. This is because *E. repens* is more closely related to barley than *A. arenaria* ssp. *australis* and *C. canescens* (both tribe Aveneae), which hosted beneficial control agents in the studies by Maciá-Vicente et al. [37]. However, none of the 24 tested endophytes showed consistently significant results in the biocontrol experiments presented here. Potentially the degree of colonisation or colonisation as an endophyte can explain the mixed success rate in this study.

Originally, around 150 fungal root endophytes were obtained from *Elymus repens* [41] and it is possible that there are successful control agents among the isolates not tested. The subset of endophyte cultures that were tested against barley diseases was selected based on their ability to sporulate. However, some sporulating cultures were not tested. It has been suggested that hundreds to thousands of microorganisms have to be screened in order to find a few beneficial ones [13,42]. However, this may not be required since several studies have shown that testing of fewer than a hundred isolates has resulted in the identification of efficient control agents [24,25,37]. Sometimes it may still be necessary to screen many strains because we have little understanding of the ecological functions of endophytes within plants, including beneficial species. In the present investigation, two isolates of *Periconia macrospinosa,* E1 and E2, originating from the same plant showed very different results in the *Fusarium* assays. This emphasises the importance of the “isolate” and illustrates why only specific isolates can be patented and made into commercial products [43]. 

It has been shown that cultivars can respond differently to the same biological control treatment. Osborne et al. [44] found that wheat cultivars showed a difference in their ability to support the colonisation of a beneficial root fungus, *Gaeumannomyces hyphopodioides*. Jørgensen et al. [45] found that their biocontrol agents could reduce disease in seven different barley cultivars. However, the magnitude of the disease reduction depended on the cultivar. There could be various reasons for this, including different abilities of BCAs to colonise the host or that the BCAs induce host plant resistance [36], differing in their response. Thus, potentially a higher number of cultivars should be tested in order to explore the full potential of a biocontrol agent. 

Biological control agents have frequently been found to give inconsistent disease control [42]. Such inconsistency is often explained by changes in the environment including, among other factors, the plant genotype, soil properties or growth conditions [46,47]. In addition, it was found that the degree of endophyte colonisation could change the lifestyle of the endophyte from antagonistic to disease facilitator [48]. In general, more consistent results would have to be shown in growth chamber experiments before trials under field conditions could be recommended.

## 4. Materials and Methods

### 4.1. Plant and Fungal Material

The highly susceptible spring barley (*Hordeum vulgare* L.) cultivar Chapeau was grown in a growth chamber under the following conditions: cycles of 16 h of light (Philips Master IL-D 36 w/865, France, 200 µmol m^−2^ s^−1^) and 8 h of darkness. Day and night temperatures were maintained at 20 °C (60% relative humidity) and 15 °C (80% relative humidity), respectively.

The pathogens used were *Fusarium culmorum* isolate 5 and *Pyrenophora teres f. teres* isolate CP2189 (both from the University of Copenhagen isolate collection). Previously, 143 endophytes had been isolated from the roots of *Elymus repens* (L.) Nevski and a subset of 24 isolates were selected for the biocontrol assays (Table 1 and Table 2; for further details see [41]). The endophytes were selected for their ability to sporulate. The endophytes were grown on PDA or MEA according to which medium they were originally isolated on (Table 2). 

All endophytes were isolated as single spore cultures after initial isolation from the host plant. Under sterile conditions, 10 µL autoclaved MiliQ water was added to an empty Petri dish. With a sterile scalpel, 4 mm^2^ of a growing fungal culture was scraped and comminuted in the 10 µL water. With a loop, the fungal material was smeared out on two water agar plates. The plates were checked under the microscope for germinating spores the following day. Germinating spores were picked out using a sterile needle prepared from a small capillary tube (Pyrex 1.3–1.5 mm × 100 mm). The single spore plug was placed on the type of medium that the original culture grew on. When the single spore culture had grown in size, plugs were frozen in 10% glycerol (v/v) at −80 °C for storage.

Endophyte identification was based on morphology and DNA sequences using three different barcoding loci. Furthermore, cultures were examined for identifiable morphological characters such as conidiophores and spores. If suitable spores were present, the length, width and additional features were measured for 100 individual spores using an Olympus BX60 light microscope. DNA was extracted from each fungal culture and ITS, LSU and TEF were amplified (for further details, see [41]). Neighbour-joining trees based on p-distance were made for each barcoding region using the software MEGA7: molecular evolutionary genetics analysis across computing platforms [49]. First trees were built with all sequences and subsequently, individual trees were built for each class of fungi separately to examine if the same number of OTUs would be determined. The sequences were clustered into OTUs using 99% sequence similarity. Consensus clusters were prepared by comparing the groups assigned by the different phylogenetic trees. If there was incongruence between OTUs among the gene regions used, the OTU determined by the most discriminating gene was chosen (best coverage and highest variability). To assign a name to the OTU clusters, all sequences were put through NCBI standard nucleotide blast (https://www.ncbi.nlm.nih.gov/, 23 November 2018) and the UNITE database (https://unite.ut.ee/ 23 November 2018, [50]). When there were discrepancies between the best hit of the different barcoding regions, the following steps were taken to allocate the taxonomic name and manage incongruence: 1) evaluate the quality of the sequence, 2) compare levels of percentage identity (only 99–100% was accepted), 3) compare spore morphology where possible and 4) give priority to the barcode hit determined by the most discriminating gene.

### 4.2. Endophyte and Pathogen Inoculation

#### 4.2.1. Inoculation of Seeds with Endophytes

A total of nine experiments were conducted (Table 1, Appendix A). Sporulating cultures of endophytes were harvested by adding 5 mL of deionised water to each Petri dish and scraping the surface with a sterilised scalpel. The suspension was filtered through a layer of cheesecloth (pore size 1×1 mm). The spore concentration was adjusted to 10^7^ spores/mL. In all the experiments, seeds were soaked in the endophyte spore suspensions (1:2 w/v) except in one experiment (exp. 5) where the spore suspension was (1:1 w/v, see Appendix A for an overview of the seed inoculations with endophytes). The seeds were placed for 10 min on a shaker at 130 rpm, except in one experiment (exp. 9) where the seeds were shaken for 24 h in the spore suspension. The seeds were dried for 30 min on filter paper in a laminar flow cabinet for all experiments except two, where seeds were dried for 2 h (exp. 9) and where seeds were dried overnight (exp. 6). Soaking in deionised water was used as a control treatment.

#### 4.2.2. Inoculation of Seeds with *Fusarium culmorum*


*Fusarium culmorum* was grown on PDA for 14–21 days. Spores were harvested as described for endophyte inoculation and the spore concentration adjusted to 1.5 × 10^6^ spores/mL. Seeds were inoculated by soaking in the spore suspension (1:1 w/v) for 30 min on a shaker at 130 rpm. In all experiments, seeds were dried for 24 h on filter paper in a laminar flow bench except for one experiment where seeds were dried for 1 h (exp. 6).

#### 4.2.3. Leaf Inoculation with *Pyrenophora teres*


*Pyrenophora teres* was grown on grass agar (filtrate of 32.5 g/L of boiled clover-rich grass fodder pills for cattle and 20 g/L agar, [45]) for 14 days. Spores were harvested by adding 5 mL of deionised water to each Petri dish and scraping the surface with a sterilised scalpel. The suspension was poured into a 50 mL tube and shaken vigorously by hand. Subsequently, the suspension was filtered through a layer of cheesecloth (pore size 1 × 1 mm). The spore concentration was adjusted to 10^3^ spores/mL. The suspension was sprayed onto the adaxial side of the leaves until run-off using a glass hand sprayer.

### 4.3. Sand Assay to Evaluate the Efficacy of Endophyte Seed Coating against Seed-Borne Fusarium culmorum 

The protocol of Jensen et al. [34] was followed. All treatments consisted of two seed dressing steps as shown in Table 3. A total of six experiments were conducted and each experiment had a water treatment, a *F. culmorum* treatment and 3–8 treatments with *F. culmorum +* endophyte, dependent on the experiment (Table 1, Appendix A). 

#### 4.3.1. Sowing and Experimental Design

Sand (grain size 0.4–0.8 mm) and tap water was mixed (3:1 v/v) and filled into strips of four plastic pots (Figure 4A). A template was used to make 1 cm deep and 2 cm in diameter wide holes in the sand. Three seeds were sown per hole and then covered with moist sand. There were four replications per treatment and each replication consisted of 12 seeds. The experiment was conducted using a fully randomized block design. Each strip of four plastic pots was placed in a saucer. Strips and saucers were placed in trays that were covered with a plastic bag in order to keep the humidity high. After 8 days, the seedlings were watered in the saucer with 50 mL of fertilizer solution (Pioner Brun, NPK 14-2-23, adjusted to pH = 6.3 with nitric acid, Azelis, Denmark) and otherwise watered every 3^rd^ day with tap water. 

#### 4.3.2. Disease Index and Disease Incidence

Disease symptoms of foot and root rot were scored 14 days after sowing. The roots were washed free of sand and the disease severity was scored using a scale from 0 to 4 (Figure 4B) described by Knudsen et al. [51] where: 0: healthy plant; 1: slightly brown root and/or coleoptile; 2: moderately brown root and coleoptile; 3: severe browning of coleoptile; 4: dead plant (non-germinated red and rotted soft seed were also included as a category 4 score). A disease index was calculated using the following Equation (1).
(1)$$\frac{0\;\times \;\mathrm{no}.\;\mathrm{of}\;\mathrm{pl}.\;\mathrm{scored}+1\;\times \;\mathrm{no}.\;\mathrm{of}\;\mathrm{pl}.\;\mathrm{scored}\;+2\;\times \;\mathrm{no}.\;\mathrm{of}\;\mathrm{pl}.\;\mathrm{scored}\;+3\;\times \;\mathrm{no}.\;\mathrm{of}\;\mathrm{pl}.\;\mathrm{scored}\;+4\;\times \;\mathrm{no}.\;\mathrm{of}\;\mathrm{pl}.\;\mathrm{scored}\;4}{\mathrm{Total}\;\mathrm{no}.\;\mathrm{of}\;\mathrm{pl}.}$$

Disease incidence was evaluated on the same plants used for the disease index. Plants were scored as either healthy (corresponding to disease index 0) or sick (corresponding to disease index 1–4). 

### 4.4. Root Colonisation by Periconia macrospinosa E1—Transformed with GFP

#### 4.4.1. *Agrobacterium tumefaciens* Mediated Transformation with GFP 

The protocol of Mullins et al. [52] was essentially followed for the *Agrobacterium*-mediated transformation of *Periconia macrospinosa* E1 as this species had demonstrated promising biocontrol properties. The strain E1 was transformed using the *A. tumefaciens* strain AGL1 containing the GFP transformation plasmid pPZP201-GG-BH (see for detailed transformation methodology). 

#### 4.4.2. Root Colonisation Experiment

Experiment 6 was set up to examine whether the transformed isolate of E1 would behave as the wild type and in order to follow the root colonisation. The transformed *Periconia macrospinosa* isolate E1gfp10 was tested in a *Fusarium* assay with a total of six treatments including “water”, “*F. culmorum*”, “E1”, “E1gfp10”, “*F. culmorum* + E1” and “*F. culmorum* + E1gfp10”. Colonisation was checked under the fluorescence microscope before germination and on the third day after sowing. Furthermore, 14 days after sowing, four root segments from plants treated with *F. culmorum,* E1 gfp10 and *F. culmorum* + E1 gfp10 were washed in tap water and placed on 1/5 PDA to check the colonisation. 

### 4.5. Net Blotch Assay to Evaluate the Efficacy of Endophyte Seed Coating against Pyrenophora teres

For the net blotch experiments, the protocol prepared by Jørgensen et al. [45] was followed with a few modifications. Thus, the plants were grown in the growth chamber and the endophyte antagonist was coated on the seeds. First, the seeds were dressed in endophyte spore suspension or water and sown. After 14 days, leaves were sprayed with the pathogen inoculum. A total of three experiments were conducted (Table 1, Appendix A).

#### 4.5.1. Sowing and Experimental Design

Rectangular pots (11.5 cm × 10 cm) were filled with Pindstup potting mix (Pindstrup Substrate no. 2, Pindstrup Mosebrug, Ryomgård, Denmark) and 10 seeds were sown in a row at a distance of 1/3 of the shortest side of the pot. There were four replications per treatment and each replication consisted of 10 seeds. At 24 h before pathogen inoculation, the 14-day-old plants had their second leaf mounted horizontally on a bent plastic plate with the adaxial side up, using two unbleached cotton strings (Figure 4C). The fixed leaves were inoculated with *Pyrenophora teres* and pots were placed in trays covered with plastic bags. Trays were kept in the dark overnight and plastic bags were opened the following day. The experiments were conducted with a fully randomized design. 

#### 4.5.2. Disease Scoring

Seven days after pathogen inoculation, images of the fixed leaves were recorded and percent lesion area was scored using the software Assess 2.0 Image Analysis Software for Plant Disease Quantification (https://my.apsnet.org/APSStore/Product-Detail.aspx?WebsiteKey=2661527A-8D44-496C-A730-8CFEB6239BE7&iProductCode=43696, American Phytopathological Society, accessed on 20 September 2022).

### 4.6. Statistical Analyses

The data were analysed with the software package R i386 3.4.3 (https://cran.r-project.org/bin/windows/base/old/3.4.3/, accessed on 12 November 2018). Linear and logistical models with the random effects “pot” and “plant” were fitted and tested against each other using ANOVA. The models were validated by checking the assumption that the observed data followed a normal or binomial distribution and that the residuals were homogenous and independent. Pairwise comparisons were made for all experiments except experiment 6, where Tukey’s range test was used. All comparisons were adjusted using Bonferroni-adjusted *p*-values at the significance level *p*
*≤* 0.05. All histograms represent means of raw data from individual experiments and error bars show the standard error of the mean (SEM). Treatments that were significantly different to the “*F. culmorum*” treated plants in the *Fusarium* assays or the “water” treatment in the net blotch assays were given an asterisk. In experiment 6, all treatments were compared to each other and means marked with different letters are significantly different.

## Figures and Tables

**Figure 1 pathogens-11-01097-f001:**
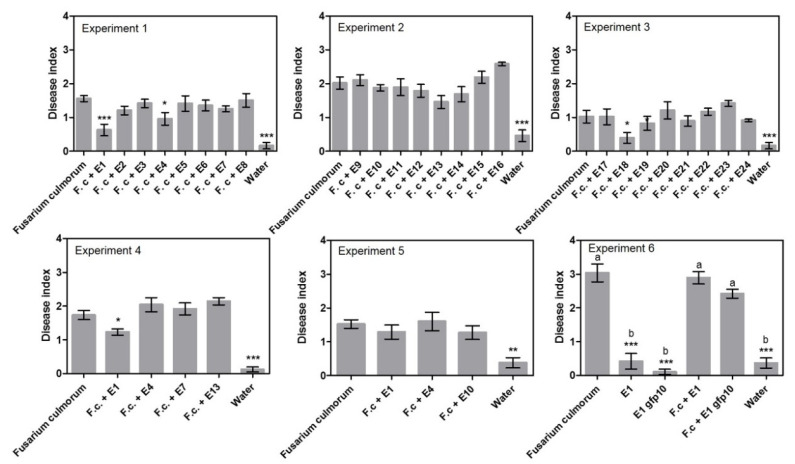
Disease index results from the sand assays testing endophytes against seed-borne *Fusarium culmorum*. Experiments 1–3 examined 24 different endophytes. Experiment 4–6 detailed investigation of selected endophytes. Disease index is given ± SEM. Each column represents the mean of 48 plants. Significant differences are compared to the “*Fusarium culmorum*” treated plants and are shown with asterisks (‘*’ 0.01 < *p* ≤ 0.05, ‘**’ 0.001 < *p* ≤ 0.01, ‘***’ *p* ≤ 0.001). In experiment 6, a gfp transformed isolate of E1 was tested to investigate if the transformation had negative effects on fitness. Therefore, all treatments were compared to each other and marked with letters to indicate dissimilarities (*p* ≤ 0.001).

**Figure 2 pathogens-11-01097-f002:**
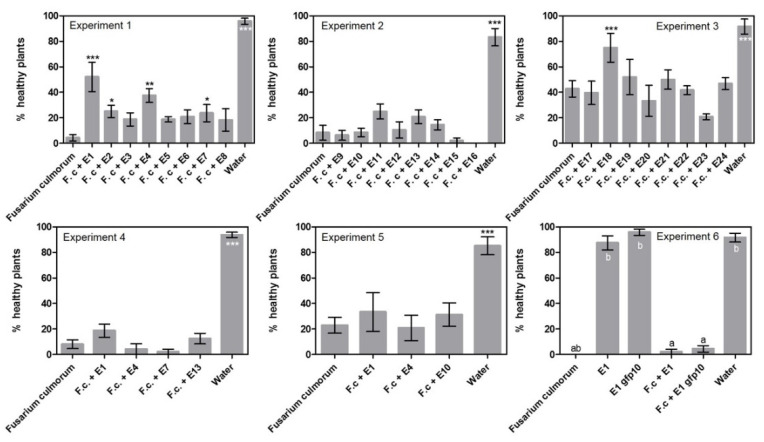
Disease incidence results from the sand assays examining endophytes against seed-borne *Fusarium culmorum*. Experiments 1–3 tested 24 different endophytes. Experiment 4–6 detailed investigation of selected endophytes. Percent healthy plants is given ± SEM. Each column represents the mean of 48 plants. Significant differences are compared to the “*Fusarium culmorum*” treated plants and are shown with asterisks (‘*’ 0.01 < *p* ≤ 0.05, ‘**’ 0.001 < *p* ≤ 0.01, ‘***’ *p* ≤ 0.001). In experiment 6, a gfp transformed isolate of E1 was tested to investigate if the transformation had negative effects on fitness. Therefore, all treatments were compared to each other and marked with letters to indicate dissimilarities (*p* ≤ 0.001).

**Figure 3 pathogens-11-01097-f003:**
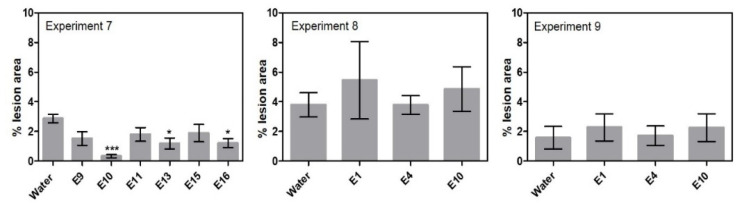
Percent lesion area results from the net blotch assays. Experiment 7 tested six isolates. Experiments 8–9 detailed investigation of selected endophytes. Results from the net blotch assays are given +/- SEM. Each column represents 40 plants. Significant differences are compared to the “water” treated plants and are shown with asterisk (‘*’ 0.01 < *p* ≤ 0.05, ‘***’ *p* ≤ 0.001).

**Figure 4 pathogens-11-01097-f004:**
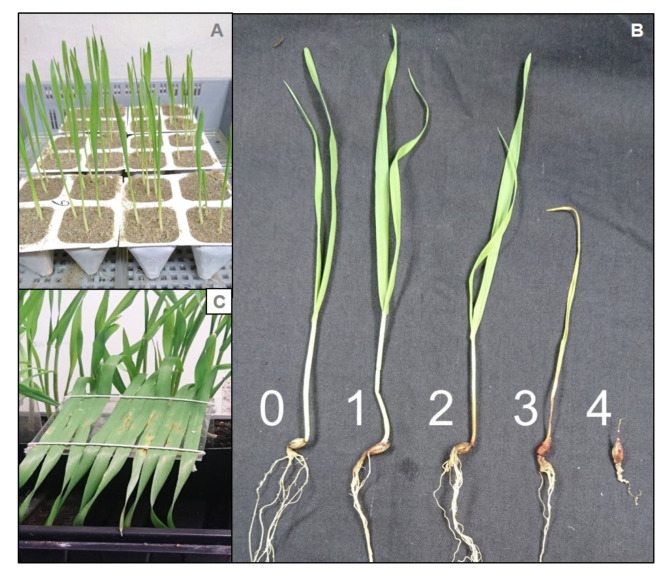
Disease symptoms in barley from *Fusarium culmorum* and *Pyrenophora teres*. (**A**) Experimental set-up of the sand assay screening candidates against *F. culmorum* showing seven-day-old plants. (**B**) Disease symptoms on a scale from 0–4 caused by *F. culmorum* in barley. (**C**) Fixed leaves of 3-week-old plants showing symptoms of *P. teres*.

**Table 1 pathogens-11-01097-t001:** Overview of efficacy of individual endophyte isolates tested for efficacy against *Fusarium culmorum* (6 experiments) and *Pyrenophora teres* (3 experiments).

Endophyte Isolate		*Fusarium culmorum*						*Pyrenophora teres*		
		Experiment Number						Experiment Number		
Label	Identification	1	2	3	4	5	6	7	8	9
E1	*Periconia macrospinosa*	+			+	–	–		–	–
E2	*Periconia macrospinosa*	+								
E3	*Slopeiomyces cylindrosporus*	–								
E4	*Epicoccum nigrum*	+			–	–			–	–
E5	*Leptodontidium* sp.	–								
E6	*Slopeiomyces cylindrosporus*	–								
E7	*Leptodontidium* sp.	+			–					
E8	*Epicoccum* sp.	–								
E9	*Periconia* sp.		–					–		
E10	Lasiosphaeriaceae sp.		–			–		+	–	–
E11	*Leptodontidium* sp.		–					–		
E12	*Leptodontidium* sp.		–							
E13	Lindgomycetaceae sp.		–		–			+		
E14	Chaetosphaeriaceae sp.		–							
E15	*Dictyochaeta siamensis*		–					–		
E16	*Leptodontidium* sp.		–					+		
E17	*Diaporthe sp.*			–						
E18	*Slopeiomyces cylindrosporus*			+						
E19	*Mycochaetophora* sp.			–						
E20	*Leptodontidium* sp.			–						
E21	Unidentified			–						
E22	*Clohesyomyces aquaticus*			–						
E23	*Ophiosphaerella* sp.			–						
E24	*Dictyochaeta siamensis*			–						
E1gfp10	*Periconia macrospinosa* gfp10						–			

A ‘+’ indicates that the endophyte was able to reduce disease symptoms significantly and ‘–‘ indicates that the endophyte had no significant effect.

**Table 2 pathogens-11-01097-t002:** Endophytes (E1–E24) tested as biological control agents in nine different experiments against the two pathogens *Fusarium culmorum* and *Pyrenophora teres*. “Plant” indicates which host plant number the endophytes were originally isolated from (1–10). The medium from which endophytes were originally isolated and the taxonomic identification, according to morphology and DNA barcoding, of the species are also given. Names of fungi follow Species Fungorum (www.indexfungorum.org, accessed on 12 December 2018).

Endophytes							
Label	Identification	Plant ^a^	Medium ^b^	Label	Identification	Plant	Medium
E1	*Periconia macrospinosa*	1	MEA	E14	Chaetosphaeriaceae sp.	10	PDA
E2	*Periconia macrospinosa*	1	MEA	E15	*Dictyochaeta siamensis*	10	MEA
E3	*Slopeiomyces cylindrosporus*	2	MEA	E16	*Leptodontidium* sp.	10	MEA
E4	*Epicoccum nigrum*	2	PDA	E17	*Diaporthe sp.*	4	MEA
E5	*Leptodontidium* sp.	3	PDA	E18	*Slopeiomyces cylindrosporus*	4	PDA
E6	*Slopeiomyces cylindrosporus*	6	PDA	E19	*Mycochaetophora* sp.	4	PDA
E7	*Leptodontidium* sp.	6	PDA	E20	*Leptodontidium* sp.	6	MEA
E8	*Epicoccum* sp.	7	PDA	E21	Unidentified	9	MEA
E9	*Periconia* sp.	2	PDA	E22	*Clohesyomyces aquaticus*	9	MEA
E10	Lasiosphaeriaceae sp.	3	PDA	E23	*Ophiosphaerella* sp.	9	MEA
E11	*Leptodontidium* sp.	3	PDA	E24	*Dictyochaeta siamensis*	7	MEA
E12	*Leptodontidium* sp.	6	MEA	E1gfp10	*Periconia macrospinosa* gfp10	-	MEA
E13	Lindgomycetaceae sp.	7	MEA				

^a^ Endophytes were isolated from ten different plants and the plants were given a number from 1–10. As an example, the endophytes E3, E4 and E9 were all isolated from *Elymus repens* plant number 2. ^b^ MEA: Malt extract agar; PDA: Potato dextrose agar.

**Table 3 pathogens-11-01097-t003:** Overview of the two seed dressing steps for the treatments used in the experiments with *Fusarium culmorum*.

Name of Treatment	First Dressing	Second Dressing
Water	Deionised water	Deionised water
*F. culmorum*	*F. culmorum*	Deionised water
*F. c* + E(number)	*F. culmorum*	Endophyte (number)

## Data Availability

Data available upon request.

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
