# Peer review of "Fungal Endophytes Isolated from Elymus repens, a Wild Relative of Barley, Have Potential for Biological Control of Fusarium culmorum and Pyrenophora teres in Barley"

_pathogens, 2022, doi:10.3390/pathogens11101097_

Round 1

Reviewer 1 Report

The study on the endophyte-mediated mitigation of critical disease in barley is valuable and informative but the experiments are not very well planned and the manuscript is non-systematically written. There are major concerns which require proper addressing

1.      The GFP based tagging experiments and the images of root colonization are inconclusive. The information provided in the supplementary file is limited and it does not reflect the root colonization image of the endophyte. And why this has been kept as supplementary rather than being the central part of the manuscript.

2.      The results depicted that E18 was highly effective and a better performer than E1 so why transformation experiments were performed on E1.

3.      Why this particular cultivar “Chapeau” was selected for this experiment. Is it reported to be resistant or susceptible for both the diseases or?

4.      Material and method section 4.1 what is the essence of giving detailed information of molecular identification of endophytes when it was not the part of this manuscript but a previous experiment.

5.      The flow of information in abstract is confusing and it should follow a systematic approach. For example, the total isolates found effective for each pathogen followed by the main isolate’s information.

6.      lines 21-23 in abstract, do not draw any conclusion.

7.      Introduction 28-34 haphazardly written. Neither barely introduction nor disease or its symptoms or economic importance are specifically addressed. At least provide one line for barley importance followed by important diseases and then introduce the diseases under study and their economic importance.

8.      Line 36: Better to write “foliar fungal pathogen” rather than “organisms infecting leaves”

9.      Several typographical mistakes and non-italic scientific mistakes must be rectified.

Reviewer 2 Report

COMMENTS TO THE AUTHORS

Dear Corresponding Author,

The manuscript numbered “pathogens-1908759” investigates the ability of some fungal endophytes of a perennial wild relative of barley on the control of foot and root barley disease (caused by F. culmorum), as well as net blotch of barley leaves (caused by P. teres) for a new application in biological control. Symptoms of these two pathogens were reduced using some endophytes as seed dressing but a widely varying degree of protection was observed between different experiments, suggesting that this approach can be correct, but it must be improved. I think you should consider your work as preliminary and you should specify this in the manuscript and the title. However, the study adds new knowledge to this very interesting topic. In general, the manuscript is well-written and adequately clear for the reader. One of my concerns is about the use of only one strain of F. culmorum and one of P. teres, for this the results could be strictly related to the “strain effect” and not to the species. I hope that in the next future you test the endophytes towards more F. culmorum and P. teres strains. The second concern is about the results that are very different in the nine experiments. In fact, the selected endophytes did not control F. culmorum in a significant way every time. This aspect limits the validation of your results. However, I provide a revision list. I hope that my suggestions and considerations can be useful to improve the paper and the clarity for the reader.

·       Please, write in the title, abstract and discussion that your investigation is a preliminary study where unfortunately the results vary in the different experiments.

·       Please, write in the abstract and discussion that your results were obtained only using one strain per species so they could be strictly related to strains and not generalized to the species.

·       Line 241: Please, add L. after Hordeum vulgare.

·       Line 245: Did you characterize P. teres? Do you know if it is P. teres f. sp. teres (causal agent of the net form of net blotch) or P. teres f sp. maculata (spot form of net blotch)?

·       Line 290: Can you specify the size of cheesecloth pores used for F. culmorum?

·       Line 309: Can you specify the size of the pore of the cheesecloth used for P. teres? It is known that filtration of P. teres conidia is not easy, due to their size.

·       In the entire manuscript write species name in italics, such as for example in lines: 69, 71, 72, 78, 79, 80, 82, 83, 85, 93, 95, 99-110, 116, 119, 298, 304 and 358.

·       Line 318: Please, control if superscript 1 is correct.

·       Line 324 and 365: Can you summarize the six/nine experiments in a Table? Because in the materials and methods the description is a little bit unclear, whereas, with Figure 2, 3 and 4 the comprehension is easier. You can build a Table with the different parameters you changed.

·       Line 331: Figure 4B in the text comes before 4A but it is unusual. I think you must re-organize Figure 4.

·       Line 378: it is not important to mention Figure 3 in materials and methods.

·       Figure 1, Figure 2, Figure 3: Can you make bigger figures?

Best regards.

Round 2

Reviewer 1 Report

The manuscript is significantly improved.

Reviewer 2 Report

Dear corresponding author,

The manuscript pathogens-1908759 now titled “Fungal endophytes isolated from Elymus repens, a wild relative of barley, have potential for biological control of Fusarium culmorum and Phyrenophora teres in barley” investigates the ability of some fungal endophytes of a wild relative of barley on the control of foot and root barley disease as well as net blotch of barley leaves for a new application in biological control. Symptoms of these two pathogens were reduced using some endophytes as seed dressing but unfortunately, a widely varying degree of protection was observed between different experiments. However, the study adds some new knowledge to this topic. For these reasons, I think the manuscript could be accepted for publication in Pathogens after the revision.

·       Line 20: Please write P. teres

·       Line 34: add ) after Ramularia collo-cygni

Best regards.